# Making a Comeback: Syphilitic Hepatitis in a Patient with Late Latent Syphilis—Case Report and Review of the Literature

**DOI:** 10.3390/pathogens11101151

**Published:** 2022-10-05

**Authors:** Alina Plesa, Liliana Gheorghe, Corina Elena Hincu, Andreea Clim, Roxana Nemteanu

**Affiliations:** 1Medical I Department, “Grigore T. Popa” University of Medicine and Pharmacy, 700115 Iasi, Romania; 2Institute of Gastroenterology and Hepatology, “Sfantul. Spiridon” University Hospital, 700111 Iasi, Romania; 3Department of Radiology, “Sfantul Spiridon” Hospital, 700111 Iasi, Romania

**Keywords:** syphilitic hepatitis, late latent syphilis, penicillin G

## Abstract

*Treponema pallidum* infection has emerged in recent years as an important community threat and burden to the health care system. Here, we report the case of a patient with cholestatic liver disease secondary to late latent syphilis. A 41 year-old male patient was referred to the clinic for assessment of an abnormal liver function panel. Ultrasound of the abdomen demonstrated an intense liver echogenicity, normal bile ducts, and no ascites. Virologic study revealed negative results for antibodies against common viral hepatitis and metabolic and autoimmune disease. The patient was tested for syphilis and a positive result was reported. The patient was diagnosed with late latent syphilis and syphilitic hepatitis and initiated benzathine penicillin at G 7.2 million units total, delivered as three doses of 2.4 million units intramuscular each at one-week intervals. He was assessed monthly and by the end of the sixth month, he had nonreactive VDRL (seroconversion), which confirmed recovery. Syphilitic hepatitis is an overlooked type of hepatitis and should be kept in mind as a differential diagnosis in an abnormal liver panel of uncertain etiology. Health care providers should be advised that higher levels of ALP may be the single landmark in cases of syphilitic hepatitis.

## 1. Introduction

*Treponema pallidum* (*T. pallidum*) infection has emerged in recent years as an important community threat and burden to the health care system. If left untreated, syphilis can progress from early to late stages, causing multiple organ damage [1]. Syphilis has diverse and subtle characteristics that make clinical diagnosis difficult, leading to a misrepresentation of true prevalence rates. Liver involvement in syphilis is a rare occurrence. Since its first description by Harn in 1943, the true prevalence of syphilitic hepatitis (SH) worldwide has remained unknown due to its unspecific and unremarkable clinical manifestations that can easily emulate other diseases [2]. Less than 10% of patients with primary and secondary syphilis can develop clinically significant hepatitis [3]. Acute liver failure or progression to liver cirrhosis is even more uncommon. Acquired SH in adult patients is seldom observed in daily practice, and the lack of definitive data published about the clinical features, diagnosis, and standard of care poses a challenge for physicians [4]. Here, we report the case of a male patient with cholestatic liver disease secondary to late latent syphilis.

## 2. Case Presentation

A 41 year-old male patient was referred to the clinic for assessment of an abnormal liver function panel. Physical examination revealed normal vital signs, painless jaundice, multiple scattered macular lesions on his torso, a post-cholecystectomy scar, and moth-eaten alopecia. The oral examination was normal. The patient recalled having completed a course of antibiotics for syphilis infection but failed to provide additional information regarding his treatment course. When questioned, he was reluctant to offer a detailed sexual history. He denied smoking, intravenous drug use, taking any other medications, receiving blood transfusions, or occupational exposure to toxins. He reported drinking occasionally. Initial investigations showed an abnormal liver panel with increased levels of serum alkaline phosphatase (ALP) of 569 IU/L, total bilirubin at 2.89 mg/dL, elevated liver enzymes, mild hypoalbuminemia, and an international normalized ratio of 1.8. A complete blood count showed normal leucocyte levels and a platelet count of 15 × 10^3^/µL. The serum electrolytes, lipids, and renal function were within the normal range. His previous history included cholecystectomy for biliary gall stones and an annual assessment of liver biochemistry that showed a persistent cholestatic syndrome. Ultrasound of the abdomen demonstrated an intense liver echogenicity, normal bile ducts, mild spleen enlargement, no ascites, and patent portal hepatic flow (Figure 1). The upper endoscopy was unremarkable. 

A dermatological assessment was performed for the atypical skin lesions, suggesting that the small macular lesions are compatible with syphilis (Figure 2). We assessed the etiology of liver cholestasis and the patient underwent computer tomography and magnetic resonance imaging, which revealed increased liver and spleen size, no bile duct obstruction, and no masses. Virologic study revealed negative results for antibodies against common viral hepatitis, HIV, and Epstein–Barr infection. Herpes simplex virus 2 and cytomegalovirus IgG were positive. Autoimmune hepatitis and primary biliary cholangitis were also assessed and excluded as possible causes of liver damage. Serum immunoglobulins, Alpha-1 antitrypsin, ceruloplasmin, and transferrin saturation were also within the normal range. After immunohistochemical and Warthin–Starry staining, the histopathology report concluded: mild mixed portal inflammation with bile duct damage and cholestasis, portal and periportal necrosis, and mild fibrosis.

Taking into consideration the patient’s personal history and the mucosal and cutaneous findings, we tested him for syphilis and found a reactive venereal disease research laboratory (VDRL), a *T. pallidum* hemagglutination assay 1:640, and a positive IgG fluorescent *T. pallidum* antibody absorbance (FTA-Abs IgG). The patient was diagnosed with late latent syphilis and SH and initiated benzathine penicillin at G 7.2 million units total, delivered as three doses of 2.4 million units intramuscular each at one-week intervals. He was assessed monthly and by the end of the sixth month, he had nonreactive VDRL (seroconversion), which confirmed recovery.

## 3. Discussions

Although syphilis is a treatable disease, with easy and inexpensive diagnostic tests, the incidence rate has been steadily increasing over the last decade [5]. According to the WHO, 17.7 million persons aged below 50 years had syphilis in 2012, with an estimated global pooled prevalence of 7.5% between 2000 and 2020 [6,7]. 

As previously reported, *T. pallidum* can affect any organ. Clinical manifestations are a consequence of local and systemic inflammatory responses caused by replication of *T. pallidum*. Neurological, genital, and cutaneous involvement have been thoroughly described in the literature [8]. However, the syphilitic infestation can have atypical presentations such as liver involvement, which can occur at any stage of the disease with a reported incidence ranging from 0.2% to 9.7% [7,9]. Secondary syphilismay develops with systemic symptoms weeks after the initial inoculation, and it is the most common stage associated with increased liver enzymes [1,10]. SH can be diagnosed by a cholestatic pattern and hypertransaminasemia with evidence of treponema infection. Mullick et al. suggested that the diagnosis is achieved when the following criteria are met: patients with a positive serology, a cholestatic liver panel in the absence of an alternative cause of hepatobiliary damage, characteristic clinical features, and who attain normalization of parameters after antibiotic therapy [10,11]. The liver enzyme pattern in SH can be characterized by a disproportionate increase in the ALP level in comparison with a modest elevation of liver enzymes and bilirubin levels, but in some cases may share hepatocellular damage, severe cholestasis, or fulminant hepatic failure [12]. Several papers have been recently published showing that SH is a growing concern and should be kept in mind as a possible cause of liver injury. Huang et al. presented the diagnostic challenges of a young patient with biopsy-proven SH [13]. Pereira et al. described a common clinical scenario of a young patient complaining of epigastric pain and nausea with worsening symptoms after conventional therapy, who returned to the emergency department with intense epigastric pain and skin lesions, to be later diagnosed with SH [14]. Similar reports by Subedi et al. and Marcos et al. are a testament to the resurgence of syphilis [12,15]. In addition, cases of acute liver failure as a manifestation of SH were reported by recent papers describing the severity of an otherwise benign infection and the challenges encountered in confirming the diagnosis [9,10]. 

The case presented here showed initial abnormal biochemical parameters with unremarkable clinical assessment, but with a positive history of syphilitic exposure. The most common causes of liver injury were systematically assessed. The patient’s liver disease was not a consequence of alcohol abuse or exposure to hepatotoxic drugs, nor did he have risk factors for viral hepatitis, which include intravenous drug use or traveling to endemic regions. The pattern of biochemical abnormalities guided further work-up of liver involvement to rule out other causes of autoimmune or metabolic disease The persistence of the cholestatic syndrome after cholecystectomy prompted the use of more sensitive imaging techniques. As transabdominal ultrasound is often non-diagnostic in identifying residual or recurrent choledocholithiasis in patients after cholecystectomy, computed tomography was performed to establish if an obstacle, either benign or malignant, lithiasic or non-lithiasic is responsible. No radiographic abnormalities were identified. Further investigations included a magnetic resonance cholangiopancreatography to assess the presence of abnormalities of the biliary tract such as strictures or dilatation of intrahepatic ducts, with normal findings. 

A definite diagnosis of SH would be the identification of spirochetes in the liver biopsy, which is an extremely rare occurrence. However, up to 50% of cases have a diagnostic biopsy after immunohistochemical stain and, thus, offer poor sensitivity and specificity for diagnosing SH [1]. Therefore, liver biopsy is a tool used to rule out other possible diagnoses rather than a confirmatory procedure for SH. Diagnostic tests for syphilis are usually affordable and accessible and are divided into nontreponemal and treponemal tests. Both types of serologic tests are required to confirm the diagnosis due to different sensitivities and specificities according to the disease stage. The most widely available tests are the microscopic VDRL and the macroscopic rapid plasma reagin (RPR) tests [16]. Our patient had a reactive VDRL titer, a positive FTA-Abs IgG, and a hemagglutination assay. The patient’s history and physical, laboratory, and abnormal findings featured SH as the most likely cause of liver cholestasis. 

Penicillin G is the treatment of choice administered parenterally irrespective of the syphilis stage. The duration and dosage of antibiotic treatment depend on the stage and clinical manifestations of syphilis [17,18]. 

The treatment of late latent and tertiary syphilis is less well defined by the current literature but generally includes penicillin. In the case presented, we opted for benzathine penicillin at G 7.2 million units total, administered as three doses of 2.4 million units intramuscular each at one-week intervals, with a positive outcome.

## 4. Conclusions

Syphilis has shown an increase in incidence, morbidity, and mortality among undiagnosed patients. *T. pallidum* infection can initially manifest as hepatic injury without any other symptom. SH is an overlooked type of hepatitis and should be kept in mind as a differential diagnosis in an abnormal liver panel of uncertain etiology. Health care providers should be advised that higher levels of ALP may be the single landmark in cases of SH. Liver enzyme levels improve after antibiotic treatment with penicillin

## Figures and Tables

**Figure 1 pathogens-11-01151-f001:**
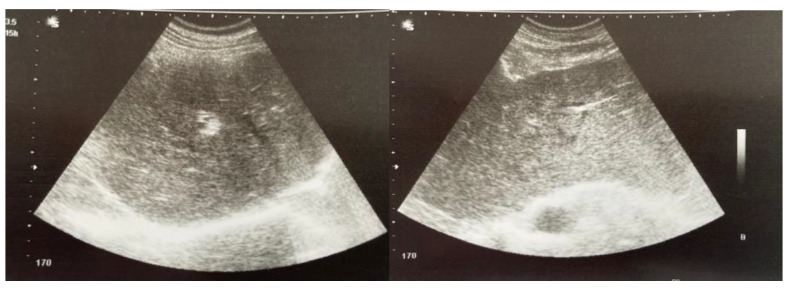
Ultrasound examination.

**Figure 2 pathogens-11-01151-f002:**
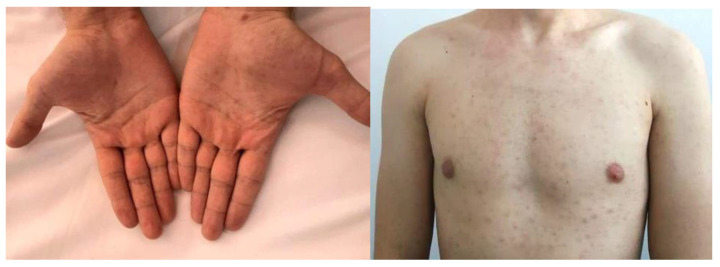
Skin examination.

## Data Availability

Not applicable.

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
