# Peer review of "Making a Comeback: Syphilitic Hepatitis in a Patient with Late Latent Syphilis—Case Report and Review of the Literature"

_pathogens, 2022, doi:10.3390/pathogens11101151_

Round 1

Reviewer 1 Report

Line 39 is incorrect.  Not every patient with untrated syphilis will develop a later stage of disease unless you consider simple positive serology as a late stage of disease. Moreover, syphilitic meningitis (a form of meningovascular syphilis can be seen with secondary disease, and even with primary disease) much earlier than 2 years.  Finally, the Oslo study showed that cardiovascular disease may not appear until 30 years after infection. 

The elevated alkaline phosphatase without substantial elevation in transaminases is quite atypical for viral hepatitis.  This diagnosis should have been made without needed for CT or MRI scanning.  Years ago, every patient who was admitted to the hospital received a test for syphilis.  This diagnosis should have been made with minimal expense

Author Response

Point-by-point Response to Reviewer’s Comments

We appreciate the reviewer for taking time to carefully review the manuscript and give detailed and constructive comments, which has greatly helped to improve this paper. Below is our point-by-point response to each comment.

Reviewer 1

  1. Line 39 is incorrect.  Not every patient with untrated syphilis will develop a later stage of disease unless you consider simple positive serology as a late stage of disease. Moreover, syphilitic meningitis (a form of meningovascular syphilis can be seen with secondary disease, and even with primary disease) much earlier than 2 years.  Finally, the Oslo study showed that cardiovascular disease may not appear until 30 years after infection. 

Response: Thank you for your observation. We have revised the paper and removed line 39, page 1.

  1. The elevated alkaline phosphatase without substantial elevation in transaminases is quite atypical for viral hepatitis.  This diagnosis should have been made without needed for CT or MRI scanning.  Years ago, every patient who was admitted to the hospital received a test for syphilis.  This diagnosis should have been made with minimal expense

Response: Thank you for your observations. Unfortunately, routine serological screening for syphilis in hospitalized patients is not a common practice in our clinical hospital, but we can confirm that a simple inexpensive serological test performed upon admission would have helped to narrow down and identify patients previously exposed to syphilis. The CT scan is useful in detecting residual choledocholithiasis in patients after cholecystectomy, yet MRCP delivers important anatomic details of the biliary tree. Therefore, we used both techniques for their different sensitivities and specificities, in a complementary manner.

Reviewer 2 Report

I have with interest this manuscript reporting a case of Syphilis hepatitis

and a review of literature.

I think that the paper is unreasonably long, and it should be shortened.

Moreover, it should be extensively revised by an English speaker.

Finally, it is unclear what the real TPHA titer was (line 129)  and what the authors wrote in line 151 (millennia?)

Author Response

Point-by-point Response to Reviewer’s Comments

We appreciate the reviewer for taking time to carefully review the manuscript and give detailed and constructive comments, which has greatly helped to improve this paper. Below is our point-by-point response to each comment.

  1. I have with interest this manuscript reporting a case of Syphilis hepatitis and a review of literature. I think that the paper is unreasonably long, and it should be shortened.

Response: Thank you for your constructive remarks. We have revised the paper and removed sentences and paragraphs from the manuscript as follows:

 Introduction: lines 47-53 and lines 63-80, page 2; Discussions:  lines 140-142 and lines 145-152, lines 163-171 page 4, lines 212-213 page 5, lines 230-234 and lines 238-241 page 6.

2. Moreover, it should be extensively revised by an English speaker.

Response: Thank you for your observation. The manuscript was revised by an expert and all changes are highlighted with track changes.

3. Finally, it is unclear what the real TPHA titer was (line 129) and what the authors wrote in line 151 (millennia?)

Response: We have revised the paper and added the patient’s TPHA titer, please see page 2, line 132.

We have removed the paragraph containing the reference mentioned – page 4, lines 145-152.

Round 2

Reviewer 1 Report

This reads too much as a chapter and not as a research paper, even for a case report.  The necessity of an ultrasound, CT and MRI of the liver needs to be explained.  It needs to be shortened extensively. 

Author Response

We appreciate the reviewer for taking time to carefully review the manuscript and give detailed and constructive comments, which has greatly helped to improve this paper. Below is our point-by-point response to each comment.

Reviewer 1:  This reads too much as a chapter and not as a research paper, even for a case report.  The necessity of an ultrasound, CT and MRI of the liver needs to be explained.  It needs to be shortened extensively. 

Response:  Thank you for your observations. We have revised the paper and removed sentences and paragraphs to shorten the length of the manuscript and reshape the paper. Please see changes in: page1 lines 30-32, and 36-46, page 2- lines 55-78, page 4-lines 145-169, page 5 lines 187-189, and lines 208-210, page 6 lines 225-226 and lines 238-245. We have also included in the Discussions section a paragraph explaining the necessity and role of the ultrasound, CT and MRI. Please see page 5 lines 210-218. The manuscript was revised by an expert and all changes are highlighted with track changes.

Reviewer 2 Report

I thank the authors for the improvement of their manuscript

Author Response

Reviewer 2: I thank the authors for the improvement of their manuscript.

We appreciate the reviewer for taking the time to carefully review the manuscript and give detailed and constructive comments, which have greatly helped to improve the quality of the paper.